# A Hybrid Surrogate Model for the Prediction of Solitary Wave Forces on the Coastal Bridge Decks

**Jinsheng Wang, Shihao Xue and Guoji Xu ***

School of Civil Engineering, Southwest Jiaotong University, Chengdu 610031, China; jinshengwangrjc@swjtu.edu.cn (J.W.); xueshihao@my.swjtu.edu.cn (S.X.)
* Correspondence: guoji.xu@swjtu.edu.cn

**Abstract:** To facilitate the establishment of the probabilistic model for quantifying the vulnerability of coastal bridges to natural hazards and support the associated risk assessment and mitigation activities, it is imperative to develop an accurate and efficient method for wave forces prediction. With the fast development of computer science, surrogate modeling techniques have been commonly used as an effective alternative to computational fluid dynamics for the establishment of a predictive model in coastal engineering. In this paper, a hybrid surrogate model is proposed for the efficient and accurate prediction of the solitary wave forces acting on coastal bridge decks. The underlying idea of the proposed method is to enhance the prediction capability of the constructed model by introducing an additional surrogate to correct the errors made by the main predictor. Specifically, the regression-type polynomial chaos expansion (PCE) is employed as the main predictor to capture the global feature of the computational model, whereas the interpolation-type Kriging is adopted to learn the local variations of the prediction error from the PCE. An engineering case is employed to validate the effectiveness of the hybrid model, and it is observed that the prediction performance (in terms of residual mean square error and correlation coefficient) of the hybrid model is superior to the optimal PCE and artificial neural network (ANN) for both horizontal and vertical wave forces, albeit the maximum PCE degrees used in the hybrid model are lower than the optimal degrees identified in the pure PCE model. Moreover, the proposed hybrid model also enables the extraction of explicit predictive equations for the parameters of interest. It is expected that the hybrid model could be extended to more complex wave conditions and structural shapes to facilitate the life-cycle structural design and analysis of coastal bridges.

**Keywords:** hybrid surrogate model; wave force prediction; coastal bridges; risk assessment; life-cycle structural design and analysis

## 1. Introduction

With the development of coastal communities and the tourist economy, the construction of coastal bridges is indispensable for establishing a complete and efficient transportation network to meet the daily commuting needs as well as to facilitate any rescue efforts after an extreme natural disaster. However, coastal bridges are often exposed to severe natural environmental conditions during their service life, and recent extreme natural events have demonstrated the vulnerability of coastal bridges to the wave forces generated by hurricanes and tsunamis, especially for the low-lying bridges that are inadequately designed for the storm surge and wave-induced forces [1–4]. Indeed, many coastal regions have sustained devastating damages to the bridges under the impact of extreme waves, e.g., more than 182 bridge spans were completely removed from their supporting structures over the gulf coast of Louisiana and Mississippi in Hurricane Katrina in 2005 and a total of 252 bridges were washed away in the 2011 Great East Japan Tsunami. The destruction of bridges may severely impact the recovery and prosperity of the coastal communities [5,6], thus it is necessary to evaluate the magnitude of wave forces and the

bridge capacity before appropriate preventive measures are taken. In this regard, a method that can accurately predict the wave forces on the bridge decks promptly is essential for the stakeholders to make critical decisions prior to the landfall of hurricanes [7]. Moreover, an effective prediction method can also facilitate the safety assessment of the bridge under a probability-based framework, e.g., structural reliability analysis [8], and enable the efficient structural analysis under the action of other extreme loads such as seismic load [9–12].

Over the last two decades, numerous research efforts have been devoted to the use of computational fluid dynamics (CFD) method for investigating the wave forces acting on bridge decks [13–17]. The lateral restraining stiffness effect on bridge deck wave interactions was studied by embedding a custom code into ANSYS Fluent [18]. Based on the smoothed particle hydrodynamics (SPH) method, the phenomenon of tsunami waves impinging on bridge superstructures was simulated [19]. Using OpenFOAM, the phenomenon of the tsunami-like wave force on box girder and T girder bridges were compared [20]. Immersed boundary method was also employed to study wave-bridge deck interactions [21]. As a time-varying dynamic system, the time-frequency characteristics of waves play an important role in the wave-structure interactions, and many scholars also have carried out relevant studies [22–25]. The influence of different wave frequencies on the motion of floating bridges was investigated [26]. It is demonstrated that the second-order difference-frequency wave loads contribute significantly to sway motion, axial force, and strong axis bending moments along floating bridges. The spectral analysis of the vertical wave forces acting on bridge decks by Fourier, wavelet, and Hilbert-Huang transform (HHT) methods were used, and then an empirical formula is proposed to predict the vertical wave forces [25]. Wavelet transforms was introduced to analyze the local characteristics of the incident waves, incline forces and transfer functions between them [27]. It is demonstrated that the nonlinear wave-structure interactions are significant for the wave components in the diffraction effect regime. Although various simulation models and analysis methods are available for the investigation of the wave forces exerted on the bridge deck, it would be time-consuming or cumbersome to obtain the prediction due to the intrinsic complicity of the bridge deck-wave interaction.

With the development of computer science and machine learning theory, the use of advanced surrogate modeling techniques in coastal engineering has drawn increasingly more attention in recent years [28–32]. By combining the M5 model tree and nonlinear regression techniques, the prediction of non-broken wave run-up on single piles is investigated in [32]. A novel model was proposed based on Extreme Learning Machine (ELM) and laboratory experiments to estimate the tsunami wave forces on coastal bridges [33]. The effects of three different machine learning techniques in predicting the wave loads on bridge decks were also compared [34]. It is proved that machine learning techniques can provide guidance for time-history prediction requirements. A new data-driven method based on the conditional Generative Adversarial Network (GAN) principle was proposed [35], through which the three-dimensional nonlinear wave loads and run-up on a fixed structure can be predicted accurately. To more efficiently predict the wave forces, the artificial neural network (ANN) is employed in [36] to establish the link between model parameters (i.e., the still-water level, wave height, and bottom elevation of the girder/superstructure) and wave forces, through which the prediction of the vertical and horizontal forces can readily be obtained in seconds. ANN was also used to quantify the loading effects with multiple surges and wave parameters [37]. Based on a wind-wave-bridge system, the effects of non-stationary winds and waves on the stochastic response of cable-stayed bridge girders were investigated using ANN [38]. It is noted, however, that the above-mentioned approaches require fine-tuning of the parameters involved in the neural network, which is a cumbersome task involving trial and error. To address this issue, a model that is easy to implement and capable of providing a predictive equation is highly desirable.

In this paper, a hybrid surrogate model based on the polynomial chaos expansions (PCE) and Kriging is proposed to establish the predictive model for the solitary wave forces acting on coastal bridge decks. The underlying idea of the proposed method is to

enhance the prediction capability of the constructed model by introducing an additional surrogate to correct the errors made by the main predictor. Specifically, this hybrid model adopts the regression-type PCE to capture the global feature of the computational model and the interpolation-type Kriging to capture the local variations of the prediction error. With the availability of the predictive model, the establishment of the probabilistic models for quantifying the vulnerability of the coastal bridges under natural hazards and the associated risk assessment can proceed easily and efficiently.

## 2. Theoretical Background

### 2.1. Polynomial Chaos Expansions

The polynomial chaos expansions (PCE) was originally proposed by Wiener to expand the stochastic process using a set of Hermite polynomials with the Gaussian random variables as the input parameters and was later generalized to account for other commonly used distributions other than Gaussian [39]. The PCE has gained its popularity for uncertainty quantification in the modern engineering community, including the ever-increasing application in the field of CFD simulations [40,41]. More recently, it has been shown that a PCE surrogate model purely trained on a data set can reach point-wise predictions with comparable accuracy to that of other machine learning models, e.g., support vector regressions and neural networks [42]. This somehow justifies the application of PCE for wave forces prediction in this study, where the data set is selected a priori.

In PCE, the simulator output (model response) is expanded onto a space spanned by a set of bases consisting of multivariate polynomials that are orthogonal to the joint probability density function (PDF) of the input variables $X$, and the model response approximated using PCE can be expressed as:

$$Y = \mathcal{M}(X) \approx \sum_{\alpha \in \mathbb{N}^n} \eta_\alpha \psi_\alpha(X) \tag{1}$$

where $\eta_\alpha$'s are the unknown coefficients to be determined and the $\alpha = (\alpha_1, \alpha_2, \ldots, \alpha_n) \in \mathbb{N}^n$ is a multidimensional index vector that indicates the components of the multivariate polynomials $\psi_\alpha(X)$, which is constructed using a tensor product of the orthogonal univariate polynomials:

$$\psi_\alpha(X) = \prod_{i=1}^{N} \phi_{\alpha_i}^i(X_i) \tag{2}$$

where $\phi_{\alpha_i}^i(X_i)$ is the orthogonal polynomial corresponding to the marginal PDF $f_{X_i}(x_i)$, satisfying $\mathbb{E}\left[\phi_m^i(X_i)\phi_k^i(X_i)\right] = 1$ if $m = k$ and 0 otherwise, for all $(m, k) \in \mathbb{N}^2$. For instance, if the variable $X_i$ follows a Gaussian distribution, $\phi_{\alpha_i}^i(X_i)$ is a set of Hermite polynomials of order $\alpha_i$, whereas Laguerre polynomials will be used for Gamma distribution. Based on this definition, the elements of the multidimensional index vector $\alpha = (\alpha_1, \alpha_2, \ldots, \alpha_n)$ of the multivariate orthonormal polynomials can also be interpreted as the degrees of the univariate polynomials and $|\alpha| = \alpha_1 + \alpha_2 + \ldots + \alpha_n$ is the degree of the corresponding multivariate polynomials.

The spectral representation of model response in Equation (1) involves an infinite number of polynomial bases, which may cause troubles in practical application. For the computational purpose, a truncation scheme is introduced for Equation (1) such that only those polynomials with total degree up to $p$ are retained, i.e., $0 \leq |\alpha| \leq p$ [43]:

$$Y = \mathcal{M}(X) \approx \mathcal{M}_{PC}(X) = \sum_{0 \leq |\alpha| \leq p} \eta_\alpha \psi_\alpha(X) = \eta^T \psi(X) \tag{3}$$

where $\eta^T = \{\eta_0, \eta_2, \ldots, \eta_{P-1}\}$ is the polynomial coefficient vector and $\psi(X) = \{\psi_{\alpha_0}(x), \psi_{\alpha_1}(x), \ldots, \psi_{\alpha_{P-1}}(x)\}^T$ is the matrix gathers all the orthonormal polynomial basis that satisfies $\{\psi_\alpha, 0 \leq |\alpha| \leq p\}$. The above formulation leads to the so-called

full PCE model, where the total number of terms involved in the expansion is given by $P = \begin{pmatrix} n+p \\ p \end{pmatrix} = \frac{(n+p)!}{n!p!}$.

Once the polynomial terms are selected, all that remains is to determine the expansion coefficients $\eta_\alpha$ using information contained in the experimental design (data set) generated from the simulator. In this study, the regression method in the category of non-intrusive approaches is employed and can be formulated as the following least-squares minimization problem [44]:

$$\hat{\eta} = \arg min \; \mathbb{E}\left[\left(\mathcal{M}(X) - \eta^T \psi(X)\right)^2\right] \tag{4}$$

Given a data set with the input vector $\mathcal{X} = \{x^1, x^2, \ldots, x^N\}^{\mathrm{T}}$ and the corresponding model responses $\mathcal{Y} = \{\mathcal{M}(x^1), \mathcal{M}(x^2), \ldots, \mathcal{M}(x^N)\}^{\mathrm{T}}$, the PCE coefficients can be estimated by solving Equation (4) using the ordinary least-square method, which gives:

$$\hat{\eta} = \left(\mathbf{\Psi}^T \mathbf{\Psi}\right)^{-1} \mathbf{\Psi}^T \mathcal{Y} \tag{5}$$

where the data matrix $\mathbf{\Psi}_{N \times P}$ is a collection of the values of polynomial basis at the experimental design points and has the following form:

$$\mathbf{\Psi} = \begin{bmatrix} \psi_{\alpha_0}(x^1) & \cdots & \psi_{\alpha_{P-1}}(x^1) \\ \vdots & \ddots & \vdots \\ \psi_{\alpha_0}(x^N) & \cdots & \psi_{\alpha_{P-1}}(x^N) \end{bmatrix} \tag{6}$$

It is noted that the size of the data set should be sufficiently large to ensure the above data matrix is well-conditioned, such that the regression problem is well-posed. Therefore, it is necessary to use an experimental design whose size $N$ is greater than the total number of terms $P$ in PCE, i.e., $P < N$. In practical applications, $N = kP$, $k \geq 2$ model evaluations are generally required to reach an approximation with sufficient accuracy.

### 2.2. Kriging

Kriging is a stochastic interpolation method where the model response is assumed to be a realization of a random function, and the Kriging model consists of a regression part and a stochastic process as follows [45]:

$$G(x) = \beta^{\mathrm{T}} f(x) + \mathcal{Z}(x) \tag{7}$$

where $f(x) = [f_1(x), f_2(x), \cdots, f_{\mathrm{m}}(x)]^{\mathrm{T}}$ is a vector of regression functions, and $\beta = [\beta_1, \beta_2, \cdots, \beta_m]^{\mathrm{T}}$ is the vector of the corresponding regression coefficients; $\mathcal{Z}(x)$ represents a Gaussian process with zero mean and the following covariance functions:

$$cov(\mathcal{Z}(x_i), \mathcal{Z}(x_j)) = \sigma_z^2 R(x_i, x_j; \theta) \tag{8}$$

where $\sigma_z^2$ is the variance of the Gaussian process; $R(x_i, x_j; \theta)$ denotes the spatial correlation function between samples $x_i$ and $x_j$, and $\theta$ is a vector of hyper-parameters to be determined. The commonly used Gaussian correlation function can be expressed as follows:

$$R(x_i, x_j; \theta) = \prod_{i=1}^{n} \exp\left(-\theta_k \left(x_{ik} - x_{jk}\right)^2\right) \tag{9}$$

where $\theta_k$ is the $k$-th correlation parameter in $\theta$; $x_{ik}$ and $x_{jk}$ are the $k$-th coordinates of samples $x_1$ and $x_2$, respectively. Given a data set with the input vector $\mathcal{X} = \{x^1, x^2, \ldots, x^N\}^{\mathrm{T}}$ and the corresponding model responses $\mathcal{Y} = \{\mathcal{M}(x^1), \mathcal{M}(x^2), \ldots, \mathcal{M}(x^N)\}^{\mathrm{T}}$, the hyper-parameters in $\theta$ can be calculated by the maximum likelihood estimation method.

Once the correlation parameters are determined, the regression coefficients $\boldsymbol{\beta} = [\beta_1, \beta_2, \cdots, \beta_m]^{\mathrm{T}}$ and the Gaussian process variance $\sigma_z^2$ can be obtained as follows:

$$\hat{\boldsymbol{\beta}} = \left(\boldsymbol{F}^{\mathrm{T}}\boldsymbol{R}^{-1}\boldsymbol{F}\right)^{-1}\boldsymbol{F}^{\mathrm{T}}\boldsymbol{R}^{-1}\mathcal{Y} \tag{10}$$

$$\hat{\sigma}_z^2 = \frac{1}{N}\left(\mathcal{Y} - \boldsymbol{F}\hat{\boldsymbol{\beta}}\right)^T \boldsymbol{R}^{-1}\left(\mathcal{Y} - \boldsymbol{F}\hat{\boldsymbol{\beta}}\right) \tag{11}$$

where $\boldsymbol{F}$ is a matrix with $F_{ij} = f_j(\boldsymbol{x}_i)$, $i = 1, \ldots, N$, $j = 1, \ldots, m$; $\boldsymbol{R}$ denotes the correlation matrix with $R_{ij} = R(\boldsymbol{x}_i, \boldsymbol{x}_j; \boldsymbol{\theta})$, $i, j = 1, \ldots, N$.

With the availability of the associated parameters, the best linear unbiased prediction of the response at a new sample point $\boldsymbol{x}^*$ can be computed as:

$$\mu_G(\boldsymbol{x}^*) = \boldsymbol{f}(\boldsymbol{x}^*)^{\mathrm{T}}\hat{\boldsymbol{\beta}} + \boldsymbol{r}(\boldsymbol{x}^*)^{\mathrm{T}}\boldsymbol{R}^{-1}\left(\mathcal{Y} - \boldsymbol{F}\hat{\boldsymbol{\beta}}\right) \tag{12}$$

$$\sigma_G^2(\boldsymbol{x}^*) = \hat{\sigma}_z^2\left(1 - \boldsymbol{r}(\boldsymbol{x}^*)^{\mathrm{T}}\boldsymbol{R}^{-1}\boldsymbol{r}(\boldsymbol{x}^*) + u(\boldsymbol{x}^*)^{\mathrm{T}}\left(\boldsymbol{F}^{\mathrm{T}}\boldsymbol{R}^{-1}\boldsymbol{F}\right)^{-1}u(\boldsymbol{x}^*)\right) \tag{13}$$

where $u(\boldsymbol{x}^*) = \boldsymbol{F}^{\mathrm{T}}\boldsymbol{R}^{-1}\boldsymbol{r}(\boldsymbol{x}^*) - \boldsymbol{f}(\boldsymbol{x}^*)$ and $\boldsymbol{r}(\boldsymbol{x}^*)$ is the vector of correlations between the new sample point $\boldsymbol{x}^*$ and the points in the training data set $\mathcal{X}$, i.e., $r_i = \boldsymbol{R}(\boldsymbol{x}^*, \boldsymbol{x}_i; \boldsymbol{\theta})$, $i = 1, \ldots, N$.

### 2.3. Proposed Hybrid Surrogate Model

In the application of surrogate modeling techniques, the relationship between the observed response $y$ and the predicted one $\hat{y}$ using a specific surrogate model can be expressed as:

$$y = \hat{y} + \varepsilon \tag{14}$$

where $\varepsilon$ is an error term that measures the deviation of the predicted value from the true one. In general, the surrogate model is first constructed from a training set and then the prediction is made directly from the model, without considering the error term during model construction and response prediction. This, however, would introduce large prediction errors if an unsuitable surrogate model is chosen for the problem at hand, especially when the given data set is small. To address this issue, a hybrid surrogate model is proposed here to establish approximating models for both structural response and prediction error. Specifically, the PCE is adopted to capture the global feature of the computational model and the Kriging model is employed to model the local variations of the prediction error, i.e.,

$$y \approx \hat{y}_{PCE} + \hat{\varepsilon}_{Kriging} \tag{15}$$

In the proposed hybrid model, the first term $\hat{y}_{PCE}$ on the right-hand side of Equaiton (15) serves as the main predictor of the structural response due to the excellent global fitting property of PCE, whereas the second term $\hat{\varepsilon}_{Kriging}$ aims to remove (reduce) the errors raised from $\hat{y}_{PCE}$. Thus, given a training data set $(\mathcal{X}, \mathcal{Y})$ for establishing the PCE, the corresponding data set for the construction of the Kriging model is $(\mathcal{X}, \mathcal{Y} - \hat{y}_{PCE})$. With the availability of the PCE and the Kriging model, the prediction of the response at a new sample point can be easily obtained from Equation (15).

In the sequel, the prediction of wave forces on a typical bridge deck-wave interaction case will be employed to investigate the applicability and validity of the proposed hybrid model.

## 3. Engineering Validation

### 3.1. Engineering Background and Data Preparation

To investigate the effectiveness of the proposed method for the prediction of wave forces, a two-dimensional bridge deck-wave interaction model as shown in Figure 1 is considered. The prototype bridge deck of this model is similar to the damaged I-10 bridge across Escambia Bay, and the solitary waves are used to represent the tsunamis and storm

surge. According to the study performed in [46], the horizontal force $F_h$ and vertical force $F_v$ can be expressed as functions of the involved parameters:

$$F_h, F_v = f(W, H, C, d, d_b, d_r, L_d, Z_c, Z_{ele}, \mu, \rho, g, \alpha) \tag{16}$$

where the wave height $H$, the wave celerity $C$ and the angle of incidence to the structure $\alpha$ are the wave variables in the model; the water depth $d$, the dynamic viscosity $\mu$ and the water density $\rho$ are the fluid-related parameters; and the structural parameters are the deck width $W$, the deck height $d_b$, the deck length $L_d$, the deck clearance $Z_c$, the rail height $d_r$ and the elevation of the bridge girder $Z_{ele}$.

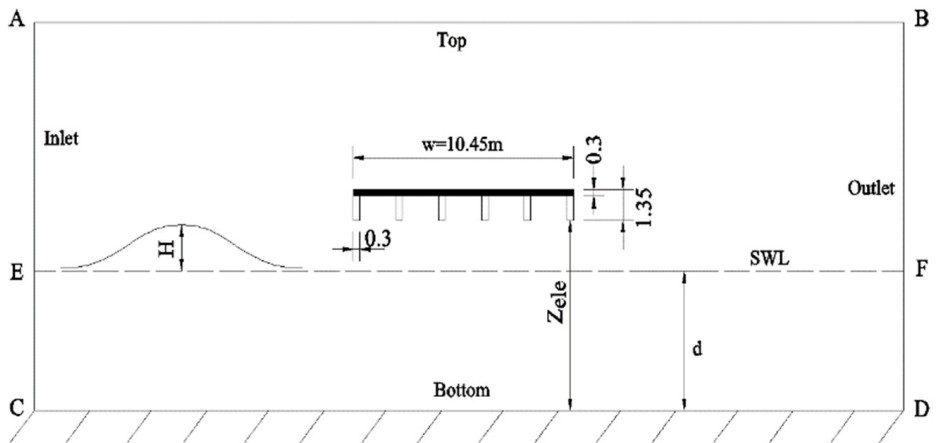

**Figure 1.** Sketch of the bridge deck-wave interaction model under solitary waves.

In this study, extensive CFD simulations are performed using ANSYS Fluent, and a total of 472 sampling pairs are generated for the construction of the hybrid surrogate model. For training surrogate models, the sampling pairs are generally composed of all the involved parameters (input) and the associated wave forces (output). However, some variables are depending on each other and/or may have a negligible effect on the evaluation of wave forces. Moreover, the required number of samples in PCE increases dramatically with the number of input parameters. Therefore, similar to the study carried out in [36], only the three critical parameters, namely the water depth $d$, the elevation of the bridge girder $Z_{ele}$, and the wave height $H$, are used as the input for establishing the prediction model. More details regarding the data preparation and the assumptions made on the bridge deck-wave interaction simulation model can be found in [15,18].

### 3.2. Surrogate Model Initiation and Assessment Metrics

The three input variables are assumed to follow a uniform distribution with a specified supporting range, as illustrated in Table 1. Thus, the normalized Legendre polynomials are used to derive the PCE, which can easily be achieved using the UQLab toolbox [44]. The degree adaptive algorithm is employed to automatically select the optimal degree of PCE according to the available data set. The Kriging module in the UQLab [45] is also employed to establish the surrogate for the prediction error of the PCE, in which the ordinary Kriging is selected for modeling the trend.

**Table 1.** Range of the considered input parameters.

| Parameter | Minimum | Maximum |
|---|---|---|
| Water depth $d$ (m) | 5 | 9.25 |
| Wave height $H$ (m) | 0.87 | 3 |
| Elevation of the bridge girder $Z_{ele}$ (m) | 2.7 | 9.6 |

The use of appropriate evaluation metrics is important for evaluating the performance of a surrogate model. The commonly used metrics include the mean absolute error (MAE), mean squared error (MSE), root mean square error (RMSE), mean relative error (MRE) and correlation coefficient (R), to name a few. Among these available metrics, MAE is less biased for higher values, yet it may not adequately reflect the performance when dealing with large error values. On the contrary, RMSE is better in terms of reflecting performance when dealing with large error values and is more useful when lower residual values are preferred. As for the R, it is a useful index that detects the linear correlation between the true and predicted values, thus can be well-suited for measuring the performance of a surrogate model. In this regard, only the RMSE and R is employed as the error metrics in the current study, and they are defined as follows:

$$\text{RMSE} = \sqrt{\frac{1}{M}\sum_{i=1}^{M}(y_i - \hat{y}_i)^2} \tag{17}$$

$$\text{R} = \frac{\sum_{i=1}^{M}(y_i - \overline{y})(\hat{y}_i - \overline{\hat{y}})}{\sqrt{\sum_{i=1}^{M}(y_i - \overline{y})^2 \sum_{i=1}^{M}(\hat{y}_i - \overline{\hat{y}})^2}} \tag{18}$$

where $M$ is the number of samples in the test data set; $y_i$ and $\hat{y}_i$ are the true response value and the response predicted by the surrogate model, respectively; $\overline{y} = 1/M \sum_{i=1}^{M} y_i$ and $\overline{\hat{y}} = 1/M \sum_{i=1}^{M} \hat{y}_i$. In the training process, the data set is split into 3 folds, where one fold is left out as the test set and the other two folds are used as the training set. Thus, three different values of RMSE (R) can be obtained after the model is trained, and the mean value of RMSE (R) is then used as the indicator of the model accuracy, i.e., a model with R close to 1 and RMSE close to 0 is deemed as the model with excellent prediction ability.

### 3.3. Results and Discussion

Given the available data set, the PCE with different maximum degrees are constructed to investigate the effects of polynomial degrees on prediction accuracy. The predicted wave forces using PCE with degrees varying from 2 to 6 and the true ones in the test data set are compared in Figures 2 and 3, and the variations of *R* and *RMSE* with the PCE degrees for horizontal and vertical wave forces prediction are listed in Tables 2 and 3, respectively. As is seen from Figure 2, the horizontal wave forces can well be predicted by the PCE with a maximum degree of 2, and increasing the maximum degree up to 5 can further improve the prediction accuracy. However, for this particular case, the PCE with a maximum degree higher than 6 does not necessarily result in a better generalization ability, in that more samples might be required to accommodate the dramatically increased number of terms in PCE. This argument is also verified from the results of assessment metrics (R and RMSE) shown in Table 2, where the R of PCE with degree 7 (R = 0.9855) is even smaller than that with degree 2 (R = 0.9943) and the RMSE of PCE with degree 7 is the largest among the investigated degrees. Although the performance of PCE for vertical wave forces prediction is slightly worse than that for horizontal wave forces prediction, as shown in Figure 3 and Table 3, the overall trend of the prediction accuracy variation is similar to that observed in Figure 2 and Table 2, except that the optimal PCE degree is 6 for vertical forces prediction.

Moreover, it is noted that the prediction performance of PCE on the horizontal wave force is better than that on the vertical force. This might be because impinging force induced by the entrapped air underneath the bridge deck makes the relationship between the input parameters and vertical wave force more complicated. A feasible way to improve the prediction accuracy on the vertical wave force is using more samples with different wave scenarios, albeit this will require more effort in data preparation.

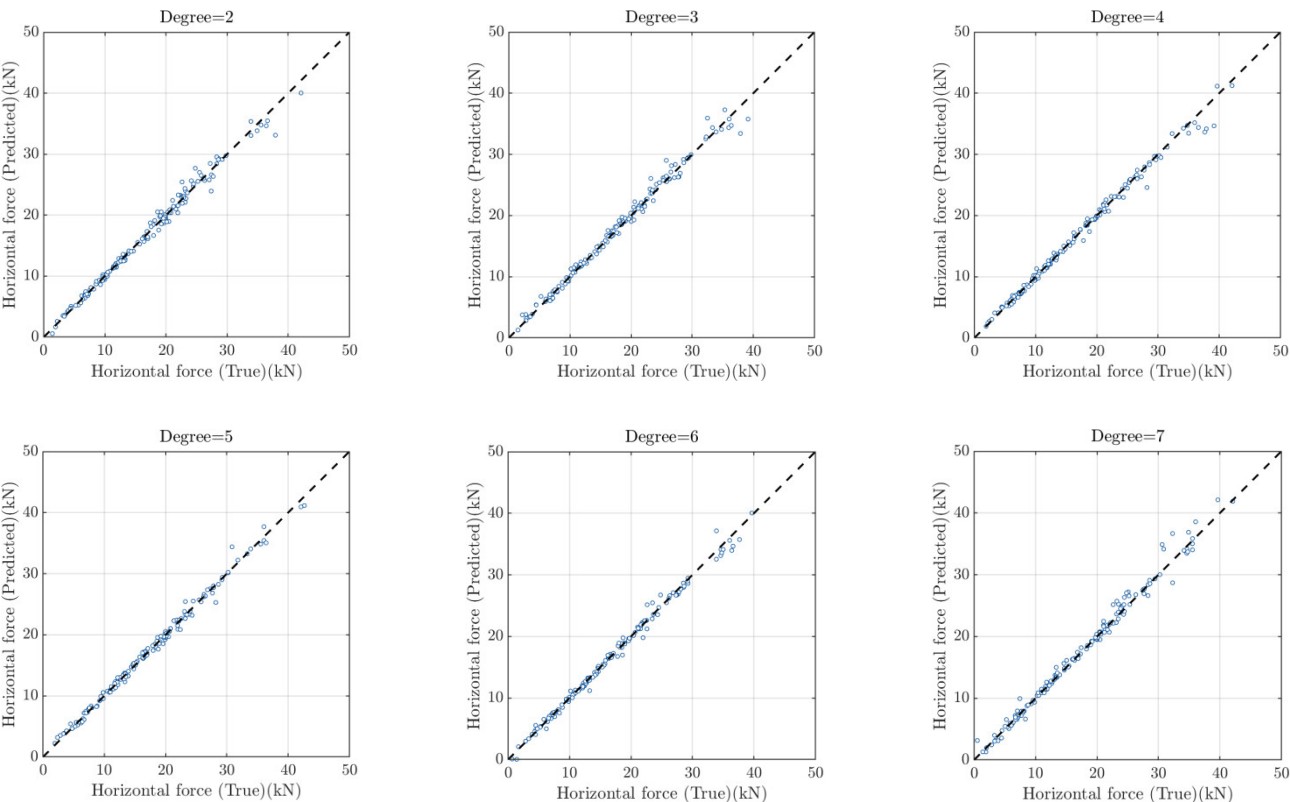

**Figure 2.** Correlation between the predicted horizontal wave forces using PCE with different degrees and the true ones in the test data set.

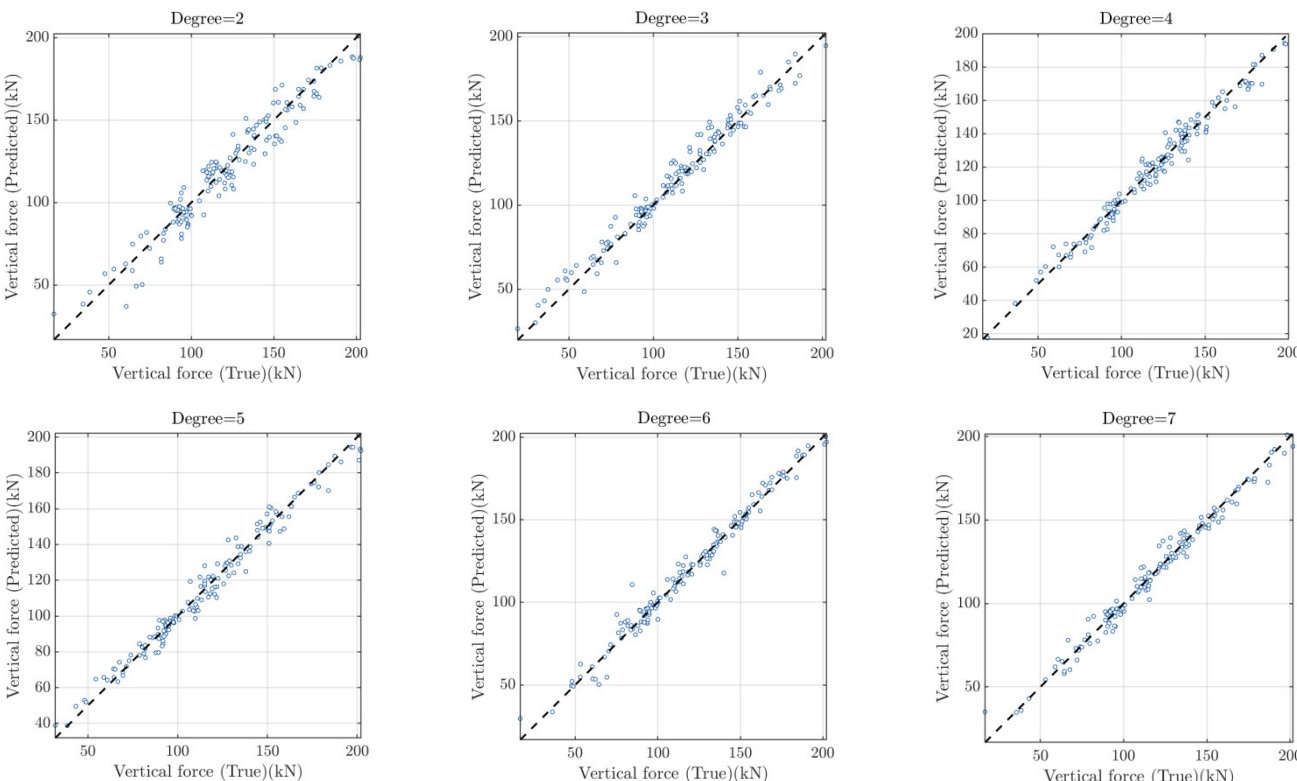

**Figure 3.** Correlation between the predicted vertical wave forces using PCE with different degrees and the true ones in the test data set.

**Table 2.** Variations of R and RMSE with the PCE degrees for horizontal wave forces prediction.

| PCE Degree | 2 | 3 | 4 | 5 | 6 | 7 |
|---|---|---|---|---|---|---|
| R | 0.9943 | 0.9945 | 0.9953 | 0.9963 | 0.9955 | 0.9855 |
| RMSE/$F_{h\_mean}$ | 5.63% | 5.60% | 5.28% | 4.58% | 5.02% | 8.19% |

**Table 3.** Variations of R and RMSE with the PCE degrees for vertical wave forces prediction.

| PCE Degree | 2 | 3 | 4 | 5 | 6 | 7 |
|---|---|---|---|---|---|---|
| R | 0.9630 | 0.9846 | 0.9838 | 0.9850 | 0.9865 | 0.9793 |
| RMSE/$F_{v\_mean}$ | 8.12% | 5.40% | 5.48% | 5.31% | 4.92% | 6.08% |

Note: $F_{h\_mean}$ and $F_{v\_mean}$ are respectively the mean value of the horizontal wave force and vertical force in the data set.

The prediction results using the proposed hybrid surrogate model is shown in Figure 4, where the optimal PCE degree for horizontal forces is found to be 2 and that for vertical forces is found to be 3. Although the maximum PCE degrees used in the hybrid model are lower than the optimal degrees identified in the pure PCE model (degree 5 for horizontal wave forces and degree 6 for vertical wave forces), the prediction performance of the hybrid model is superior to the optimal PCE for both horizontal and vertical wave forces. Specifically, for the horizontal wave forces prediction, the R and RMSE of the hybrid model are found to be 0.9975 and 3.70%, respectively; and these two values are found to be 0.9910 and 4.00% for the vertical wave forces prediction. Moreover, the results of the optimal ANN reported in [32] are also illustrated here for comparison purposes, as shown in Figure 5. Obviously, the proposed hybrid model exhibits better performance than the optimal ANN for horizontal wave forces prediction, and comparable accuracy is achieved in predicting the vertical forces for both models. Overall, the results verify the effectiveness of the error correction term in the proposed hybrid model to reduce the prediction error made by the PCE. In addition, it should be noted that the proposed hybrid model is easily implementable, without needing to tune numerous hyper-parameters and model structures as required in the ANN.

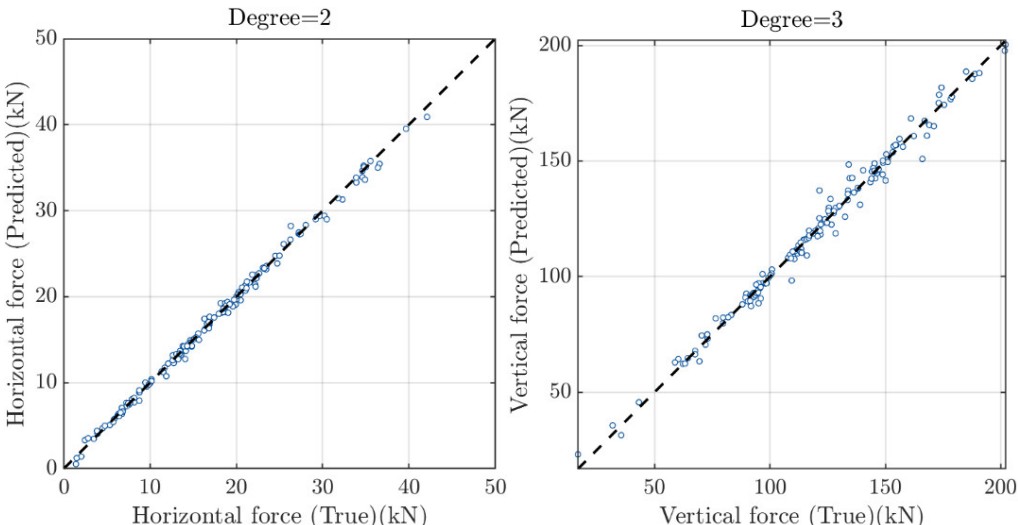

**Figure 4.** Correlation between the predicted wave forces using the proposed hybrid surrogate model and the true ones in the test data set.

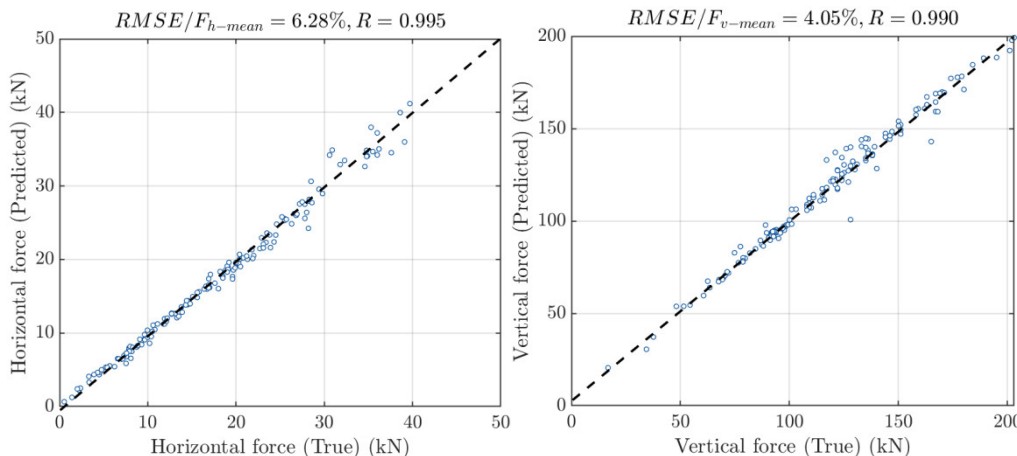

**Figure 5.** Correlation between the predicted wave forces using the optimal ANN and the true ones in the test data set.

With the availability of the trained hybrid model, the predictive equations for the horizontal wave forces and vertical forces are obtained as follows:

$$F_h = \sum_{\boldsymbol{\alpha} \in \mathbb{N}^3, |\boldsymbol{\alpha}| \leq 2} \theta^h_{\boldsymbol{\alpha}} \Psi^h_{\boldsymbol{\alpha}}(d, H, Z_{ele}) + \hat{\varepsilon}^h_{Kriging} = \boldsymbol{\theta}^T_h \boldsymbol{\psi}_h + \hat{\varepsilon}^h_{Kriging}$$

$$F_v = \sum_{\boldsymbol{\alpha} \in \mathbb{N}^3, |\boldsymbol{\alpha}| \leq 3} \theta^v_{\boldsymbol{\alpha}} \Psi^v_{\boldsymbol{\alpha}}(d, H, Z_{ele}) + \hat{\varepsilon}^v_{Kriging} = \boldsymbol{\theta}^T_v \boldsymbol{\psi}_v + \hat{\varepsilon}^v_{Kriging}$$

where

$\theta^T_h = [12.9545 \; 9.3748 - 3.0658 - 0.1475 \; 0.6793 - 5.6262 - 2.3923 \; 0.4634 - 0.8147 \; 8.4055];$

$\theta^T_v = [42.9168 \; 28.9734 - 154.2690 \; 122.7512 \; 1.0402 - 58.9573 \; 25.5081 \; 9.7797 - 8.3700$
$88.1522 \; 0.8524 - 54.7363 \; 11.3018 - 3.4960 \; 1.0044 \; 1.2990 \; 114.0724 - 0.0104 - 68.5711 \; 0.5590];$

$\hat{\varepsilon}^h_{Kriging} = -0.076 + r(x^*)^T R^{-1} \left( \mathcal{Y}_h - \theta^T_h \psi_h + 0.076 \right);$

$\hat{\varepsilon}^v_{Kriging} = 7.1368 + r(x^*)^T R^{-1} \left( \mathcal{Y}_v - \theta^T_v \psi_v + 7.1368 \right);$

$\psi_h = \left[ \psi_{(0,0,0)}, \; \psi_{(0,0,1)}, \; \psi_{(0,1,0)}, \; \psi_{(1,0,0)}, \; \psi_{(0,0,2)}, \; \psi_{(0,2,0)}, \; \psi_{(2,0,0)}, \; \psi_{(0,1,1)}, \; \psi_{(1,0,1)}, \; \psi_{(1,1,0)} \right]^T.$

$\psi_v = \left[ \psi^T_h, \psi_{(0,0,3)}, \; \psi_{(0,3,0)}, \; \psi_{(3,0,0)}, \; \psi_{(0,1,2)}, \; \psi_{(0,2,1)}, \; \psi_{(1,0,2)}, \; \psi_{(1,2,0)}, \; \psi_{(2,0,1)}, \; \psi_{(2,1,0)}, \; \psi_{(1,1,1)} \right]^T.$

$\psi_{(0,0,0)} = 1;$

$\psi_{(0,0,1)} = \frac{X_3}{\frac{1}{\sqrt{3}}}, \; \psi_{(0,1,0)} = \frac{X_2}{\frac{1}{\sqrt{3}}}, \; \psi_{(1,0,0)} = \frac{X_1}{\frac{1}{\sqrt{3}}};$

$\psi_{(0,0,2)} = \frac{\frac{1}{2}\left( 3X_3^2 - 1 \right)}{\frac{1}{\sqrt{5}}}, \; \psi_{(0,2,0)} = \frac{\frac{1}{2}\left( 3X_2^2 - 1 \right)}{\frac{1}{\sqrt{5}}}, \; \psi_{(2,0,0)} = \frac{\frac{1}{2}\left( 3X_1^2 - 1 \right)}{\frac{1}{\sqrt{5}}};$

$\psi_{(0,1,1)} = \left( \frac{X_2}{\frac{1}{\sqrt{3}}} \right) \left( \frac{X_3}{\frac{1}{\sqrt{3}}} \right), \; \psi_{(1,0,1)} = \left( \frac{X_1}{\frac{1}{\sqrt{3}}} \right) \left( \frac{X_3}{\frac{1}{\sqrt{3}}} \right), \; \psi_{(1,1,0)} = \left( \frac{X_1}{\frac{1}{\sqrt{3}}} \right) \left( \frac{X_2}{\frac{1}{\sqrt{3}}} \right);$

$\psi_{(0,0,3)} = \frac{\frac{1}{2}\left( 5X_3^3 - 3X_3 \right)}{\frac{1}{\sqrt{7}}}, \; \psi_{(0,3,0)} = \frac{\frac{1}{2}\left( 5X_2^3 - 3X_2 \right)}{\frac{1}{\sqrt{7}}}, \; \psi_{(3,0,0)} = \frac{\frac{1}{2}\left( 5X_1^3 - 3X_1 \right)}{\frac{1}{\sqrt{7}}};$

$\psi_{(0,1,2)} = \left( \frac{X_2}{\frac{1}{\sqrt{3}}} \right) \left( \frac{\frac{1}{2}\left( 3X_3^2 - 1 \right)}{\frac{1}{\sqrt{5}}} \right), \; \psi_{(0,2,1)} = \left( \frac{\frac{1}{2}\left( 3X_2^2 - 1 \right)}{\frac{1}{\sqrt{5}}} \right) \left( \frac{X_3}{\frac{1}{\sqrt{3}}} \right), \; \psi_{(1,0,2)} = \left( \frac{X_1}{\frac{1}{\sqrt{3}}} \right) \left( \frac{\frac{1}{2}\left( 3X_3^2 - 1 \right)}{\frac{1}{\sqrt{5}}} \right);$

$\psi_{(1,2,0)} = \left( \frac{X_1}{\frac{1}{\sqrt{3}}} \right) \left( \frac{\frac{1}{2}\left( 3X_2^2 - 1 \right)}{\frac{1}{\sqrt{5}}} \right), \; \psi_{(2,0,1)} = \left( \frac{\frac{1}{2}\left( 3X_1^2 - 1 \right)}{\frac{1}{\sqrt{5}}} \right) \left( \frac{X_3}{\frac{1}{\sqrt{3}}} \right), \; \psi_{(2,1,0)} = \left( \frac{\frac{1}{2}\left( 3X_1^2 - 1 \right)}{\frac{1}{\sqrt{5}}} \right) \left( \frac{X_2}{\frac{1}{\sqrt{3}}} \right);$

$\psi_{(1,1,1)} = \left( \frac{X_1}{\frac{1}{\sqrt{3}}} \right) \left( \frac{X_2}{\frac{1}{\sqrt{3}}} \right) \left( \frac{X_3}{\frac{1}{\sqrt{3}}} \right).$

$X_1 = \frac{2}{9.25 - 5}(d - 5) - 1 = 0.4706 * d - 3.353;$

$X_2 = \frac{2}{9.6 - 2.7}(Z_{ele} - 2.7) - 1 = 0.2899 * Z_{ele} - 1.7826;$

$$X_3 = \frac{2}{3-0.87}(H - 0.87) - 1 = 0.939 * H - 1.8169;$$

## 4. Conclusions

To facilitate the establishment of the probabilistic model for quantifying the vulnerability of coastal bridges to natural hazards and support the associated risk assessment and mitigation activities, a hybrid surrogate model is proposed for efficient and accurate prediction of the solitary wave forces acting on coastal bridge decks and the corresponding predictive equations are obtained from the trained model. Unlike traditional surrogate models, this hybrid model includes an error correction term to reduce the prediction error from the main predictor. Specifically, the regression-type polynomial chaos expansion (PCE) is employed as the main predictor to capture the global feature of the computational model, whereas the interpolation-type Kriging is adopted to capture the local variations of the prediction error from the PCE. The prediction of wave forces on a typical bridge deck-wave interaction model is carried out and compared with other methods to demonstrate the effectiveness of the hybrid surrogate model. According to the obtained results, the following conclusions can be drawn:

1.  The comparison among the predictive results of the PCE, the hybrid model, and those from the ANN indicates the enhanced performance of the proposed method. In other words, this hybrid model can capture the underlying physical complexities in the bridge deck-wave interaction, and can thus be used to replace the original time-consuming CFD models for the wave forces prediction and the associated life-cycle-based probabilistic modeling.
2.  The use of PCE and Kriging in this study offers several desirable advantages, e.g., the number of tuning parameters can be relatively small. In other words, only the maximum polynomial degree $p$ needs to be tuned in the PCE, enabling the easy implementation of this approach. Moreover, the time required to establish the PCE and Kriging is only a few seconds on a standard laptop, making the prediction of wave forces rather efficient. These features distinguish the proposed hybrid model from other well-known machine learning approaches such as ANNs, which are known to be highly sensitive to their hyper-parameters and require an appropriate and generally cumbersome calibration procedure.
3.  The prediction performance of PCE on the horizontal wave force is better than that on the vertical force. This might be because impinging force induced by the entrapped air underneath the bridge deck makes the relationship between the input parameters and vertical wave force more complicated. A feasible way to improve the prediction accuracy on the vertical wave force is using more samples with different wave scenarios, albeit this will require more effort in data preparation.

The limitations of the current study and future work are as follows:

1.  In the proposed hybrid model, only the PCE is used as the main predictor. However, this choice may not be appropriate when the number of training data is small, especially for engineering cases with many input parameters. Thus, the use of other effective surrogate models (e.g., support vector regression, radial basis function) or ensemble models as the main predictor may further enhance the applicability of the hybrid model.
2.  Since the training data in the engineering case is predefined, the number of samples in the data set might be too large or too small for the problem at hand, which could jeopardize the overall performance of the established surrogate model. Thus, the use of an adaptive algorithm that sequentially adds training samples to refine the surrogate model is a topic worth further exploring.

**Author Contributions:** Conceptualization, J.W.; methodology, J.W. and S.X.; software, J.W.; validation, J.W., S.X. and G.X.; formal analysis, J.W. and S.X.; investigation, J.W. and S.X.; resources, G.X.; data curation, J.W. and G.X.; writing—original draft preparation, J.W. and S.X.; writing—review and

editing, G.X.; visualization, J.W.; supervision, G.X.; project administration, G.X.; funding acquisition, G.X. All authors have read and agreed to the published version of the manuscript.

**Funding:** This research was funded by NSFC, grant number 52078425.

**Institutional Review Board Statement:** Not applicable.

**Informed Consent Statement:** Not applicable.

**Data Availability Statement:** The details of the proposed methodology and of the specific values of the parameters considered have been provided in the paper. Hence, we are confident that the results can be reproduced. Readers interested in the source code are encouraged to contact the authors by email.

**Acknowledgments:** Constructive comments from the anonymous reviewers are highly acknowledged.

**Conflicts of Interest:** The authors declare no conflict of interest. The funders had no role in the design of the study; in the collection, analyses, or interpretation of data; in the writing of the manuscript, or in the decision to publish the results.

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
