# Peer review of "A Hybrid Surrogate Model for the Prediction of Solitary Wave Forces on the Coastal Bridge Decks"

_infrastructures, doi:10.3390/infrastructures6120170_

Round 1

Reviewer 1 Report

The paper "A hybrid surrogate model for the prediction of solitary wave forces on the coastal bridge decks" reports and interesting work concerning the evaluation of the vulnerability of costal bridges to natural hazards. The topic of the article is very current and of great interest to the scientific community. The manuscript is generally well organized in the different Sections and the method proposed is clearly explained in the text. For these reasons it is opinion of this reviewer that the paper can be considered for the publication in Infrastructures Journal, after the following improvements:

  • in the Introduction consider the various problems that afflict the existing bridges under the different exceptional loads (i.e. the seismic action as reported in 10.1016/j.engfailanal.2020.104727, 3390/app10010017, 10.3390/INFRASTRUCTURES5060052, 10.1061/(ASCE)ST.1943-541X.0000220)
  • in the Conclusions better highlight the original aspects of the work and future developments.

Author Response

Thanks for your constructive comments and we have carefully addressed the comments. Please see the response letter as attached for details.

Reviewer 2 Report

The manuscript "A hybrid surrogate model for the prediction of solitary wave forces on the coastal bridge decks" is relevant and adheres to the scope of this journal, however the authors should make further revisions for its possible acceptance:

a) The abstract needs to more clearly highlight the main objective of this research, this is important to readers, in addition to better highlighting the results and main conclusions;
b) The general format of the manuscript does not comply with the MDPI standards, please review this item please;
c) Discussions are limited, there are few references in general to the manuscript, which indicates a need for further details by the authors;
d) Figs. 2, 3 and similar ones need to be reformulated so that readers can read it completely;
e) The conclusion item must be placed in topics;
f) Consider indicating a topic to suggest future work and new perspectives.

Author Response

Thanks for your constructive comments and we have carefully addressed these comments. Please see the attached response letter for details.

Reviewer 3 Report

The authors presented a hybrid surrogate model for the prediction of wave forces on coasted bridge decks. The manuscript is eloquently written and I enjoyed reading it. The paper is very clearly presented, however, the following clarifications are required to further improve the quality of the article. 

  1. It is suggested that if possible, all the figures should be redrawn to improve the pixels and clarity. The axis labels are barely readable. 
  2.  A short explanation should be added why RMSE and R are used as performance evaluators. Other parameters that can be used to evaluate performance should also be added. 
  3. In the introduction section, the authors have explained, the earlier work based on ML models and neural networks, however, the authors left time-frequency analysis as one of the major themes of prediction of forces on bridge decks due to coastal waves. It is suggested to include a short paragraph on it.
  4. The following papers should be added

(a) Chen, X., Chen, Z., Xu, G., Zhuo, X., & Deng, Q. (2021). Review of wave forces on bridge decks with experimental and numerical methods. Adv. Bridge Eng., 2(1), 1–24.

(b) Sony, S., & Sadhu, A. (2020). Synchrosqueezing transform-based identification of time-varying structural systems using multi-sensor data. J. Sound Vib., 486, 115576.

(c) Moideen, R., Ranjan Behera, M., Kamath, A., & Bihs, H. (2019). Effect of Girder Spacing and Depth on the Solitary Wave Impact on Coastal Bridge Deck for Different Airgaps. J. Mar. Sci. Eng., 7(5), 140.

Author Response

Thanks for the constructive comments and we have carefully revised the manuscript accordingly. Please see the response letter as attached for details.

Round 2

Reviewer 2 Report

The manuscript can be accepted for publication in this journal.

Reviewer 3 Report

The authors have addressed the comments satisfactorily. I recommend the paper for publication.